# Knot Factories with Helical Geometry Enhance Knotting and Induce Handedness to Knots

**DOI:** 10.3390/polym14194201

**Published:** 2022-10-07

**Authors:** Renáta Rusková, Dušan Račko

**Affiliations:** Polymer Institute, Slovak Academy of Sciences, Dúbravská Cesta 9, 845 41 Bratislava, Slovakia

**Keywords:** polymer, DNA, chirality, topology, knots, molecular dynamics, compression, nano-channels, helical, knot factory

## Abstract

We performed molecular dynamics simulations of DNA polymer chains confined in helical nano-channels under compression in order to explore the potential of knot-factories with helical geometry to produce knots with a preferred handedness. In our simulations, we explore mutual effect of the confinement strength and compressive forces in a range covering weak, intermediate and strong confinement together with weak and strong compressive forces. The results find that while the common metrics of polymer chain in cylindrical and helical channels are very similar, the DNA in helical channels exhibits greatly different topology in terms of chain knottedness, writhe and handedness of knots. The results show that knots with a preferred chirality in terms of average writhe can be produced by using channels with a chosen handedness.

## 1. Introduction

Knot factories are nano-technological devices designed to induce self-entanglements to long molecules and altering topological state of the molecule toward more knotted states [1,2,3,4,5,6]. The topological state of molecules greatly affects physical and bio-physical properties of molecules and has important technological and biological implications. While there do exist synthetic methods that allow preparing relatively small molecular knots by template reactions [7], larger knots are usually artificially produced by inducing self-entanglements to the chain by physical mechanisms. In the knot factories, the knots are produced as a result of interplay between physical properties of the molecule, confinement strength and external force perturbing the chain.

The long molecules, synonymously known as polymers, are molecules consisting of repeating building units called monomers [8]. Chemical composition and bonding of the monomers in the polymeric chain determines fundamental physical properties, such as polymer chain’s width, length and bending rigidity [9]. These determine the polymer chain’s physical properties on large scales, known as polymer chain metrics, measured by radius of gyration and molecular extension [10], that would further translate to macroscopic properties exhibited by materials, such as polymer viscosity, elasticity, etc. [11].

Polymer metrics such as the gyration radii, *R*_g_, and chain extension, *R*, are greatly affected if polymeric chains are constrained by confinement in narrow spaces [12,13]. The effect of confinement arises from competition between geometric measures of confining spaces and metrics of a polymer chain in unperturbed state. In the case of polymers, the effect of confinement is expressed as confinement strength given as a ratio of polymer stiffness, *P*, and a geometric characteristic of confinement such as width or diameter of confining channels, *D*. Here, several regimes of confinement strength on the scaling of polymer metrics have been described. For very large, basically, open spaces characterized by *D* >> *P*, scaling of polymer properties such as the gyration radii followed de Gennes regime [14], which describes polymer as a succession of blobs of a diameter *D*. In such an extremely weak confinement, the metric properties of polymer chains are unconstrained and the same as in the case of unperturbed polymers in the bulk. On the other hand, if the geometric dimensions of confining spaces *D* are much smaller than the characteristic dimension of polymer *P*, *D* << *P*, the scaling properties of polymers are described by Odijk’s deflective regime where the polymer segments are stretched as the confinement prevents the polymer chain from folding on itself [15]. In between these two boundary regimes, where the ratio between polymer rigidity and geometric dimensions of confinement *D*/*P*~1, existence of multiple intermediate and transition regions have been predicted [16], such as extended de Gennes regime or back-folding Odijk’s regimes [17,18]. Polymers in a confined state can be found in nano-technological applications such as polymer nano-composites, where the polymers interact with structurally rigid nano-particles of a filler, or polymers are squeezed between layered nano-clays, but the confined systems include also polymer blends with large difference of *T*_g_ [19,20,21]. In the case of DNA, that is a bio-polymer and vital molecule for life whose production greatly exceeds man-made polymers [22], the DNA is found in confinement naturally when it is tightly packed in cells nuclei or viral capsids. In addition, in the case of the DNA, the confinement is encountered in nanotechnological devices used in single molecular experiments to study DNA, where the confinement is used to keep DNA stretched and exposed to sensors [5,23].

Yet another type of constraint is realized by external fields that can exert tension or compression to the polymer chain. While extension of polymers is more common and it is realized experimentally for example by AFM experiments [24] or mechanical tweezers [25], compression of polymers is much less understood as it also requires presence of some type of geometrical confinement. When a combination of the two constraints is imposed on polymer, the polymer chain will experience counteraction of two forces, i.e., compressive force responsible for collapsing the polymer chain, having to compete with the confinement strength, *D*/*P*, inducing polymer pre-stretching. Upon compression the polymer’s packing and concentration increases and polymer becomes more self-entangled, leading to increased complexity of chain’s topological states characterized as polymer knotting. Compression of polymers in confined spaces encompasses a range of important real-life situations. Compression of polymer chains in nanochannels is commonly experienced in in nanotechnological devices, but it plays also vital role in living organisms, where the DNA is compressed in narrow spaces of cell nuclei, bacteriophages or viral capsids, while the DNA also keeps performing its biological functions that involve pulling and extrusion of DNA by molecular machinery [26,27,28,29].

The first experimental work investigating single molecule of DNA under compression in nano-channels used electrical fields [30]. Later experimental works used more sophisticated designs employing a gasket i.e., a nanoparticle driven by mechanical tweezers and investigated DNA subjected to a cyclic sequence of compressions followed by free expansion periods [3,4]. Alternatively, the compression of DNA chains in experiments has also been realized by employing flow of media to polymer chains driven through convergent/divergent channels [31]. When it comes to single molecular experiments, a great deal of insight can be achieved by employing molecular simulations that over decades evolved into an indispensable tool to classical experiments or theoretical approach. Compression of polymer chains under confinement was studied by means of Monte Carlo (MC) [32,33,34] and Molecular Dynamics (MD) simulations [31,35,36,37,38,39], in channels with square [32,39], cylindrical [33,34,35,36,37,38] or structured channels [31,40], when the compressive force was applied by pulling on distant ends of the confined polymer [32,33,35], by using a piston compression similar to the gasket used in experiments [33,34,36,37,39], or by flow of media [31,40]. In a combined theoretical and computer simulations MD study, force-induced deformations of a self-avoiding chain confined inside a cylindrical cavity were probed by means of molecular dynamics simulations and scaling laws for chain extension were revealed for strong (*R*/*R*_0_ << 1) and weak (*R*/*R*_0_ > 0.5) compression regimes characterized by the ratio of the chain extension *R* in compressed and unperturbed states, *R*_0_ [38]. The MC simulations were used to study force displacement relations with conformational and free energy changes during compression of linear [32,33] and ring polymers [34] and indicated different energy costs of confinement strength and external force on polymer extension with potential implications for experiments in nanofluidic chips. Computer simulations of compressed confined polymer chains employing different mechanisms of compression performed whether by pulling the distant parts of polymer chain against each other [35], piston compression [37,39] or compression by flux of media [31,40] repeatedly reported a new topological phenomenon of chiral helical structures formation. Extensive computer simulations of compression allowed exploring parameter space and revealed complex topological behaviour of emerging self-entanglements with a promise of producing knots with a desired topology by controlling compression “waiting time” and applied compressive force [36].

The aim of this paper is to explore DNA polymer chain under two constraints realized by confinement of the polymer into nanochannels and applying a compressive force. We explore polymers in a range of confinement strengths, characterized by *D*/*P* = 0.5, 1.0 and 2.0, that were already explored by MC simulations in existing works addressing polymers in square and cylindrical nanochannels, however without addressing the topological states [32,33,34]. Moreover, in our simulations, we focus on polymers confined in channels with helical geometry and we develop an implicit model of helical constraints for molecular simulations. The helical geometry has intriguing aspects: first of all, the standing models of cylindrical and square channels can be too idealized, and it was pointed out earlier that during fabrication of nano-channels irregularities and defects are created [41], that are responsible for enhanced knotting, hence grooves in the helical channels could simulate such irregularities. Moreover, in bacterial and cell nuclei, the confining spaces formed by the crowded environment would be even more shifted from idealized geometries, possibly towards the helical one. This is because the DNA in vivo is often found in highly supercoiled state due to the essential role of supercoiling on vital functions of DNA, and an environment formed from crowded supercoils would give rise to chiral confining geometries more likely to be approximated by helical shape [37,42,43]. Consequently, in our previous work [44], we have shown that helical channels exhibit stereospecific sensitivity on chiral superstructures, i.e., polymeric knot. The stereospecific effect was found to be very sensitive even for very small helical radii starting from *R*_H_ = 0.03 *R*_ch_. Therefore, in the current work we also explore the aspect of different application of the helical channels, i.e., whether and at which extent of compression in helical channels similar to the experimental setup of knot factories can produce knots with given chirality induced by handedness of the helical channel.

## 2. Materials and Methods

### 2.1. The Model of DNA

As a model system we used bio-polymer DNA that is modelled as a discretized semi-flexible worm-like chain consisting of *N* = 300 beads interacting via intra-molecular covalent interactions and inter-molecular non-bonded excluded volume interactions. One bead represents a coarse-grained portion of DNA corresponding to 8.5 basepairs with a diameter given as 2.5 nm that represents also the width of the hydrated DNA chain [45]. The total physical length of the DNA *L* = *Nσ* is 2.55 kbp what corresponds to the size encountered biologically in small plasmids. The covalent bonds are modelled by two bonded interactions. At first, bond stretching is modelled by harmonic potential function in the form *U*_s_(*r*) = *k*_s_(*r* − *r*_0_)^2^, where the constant *k*_s_ represents penalty against stretching of the bond and *r*_0_ is an equilibrium distance that is set to 1 *σ*. We limited the stretching of the bonds in order to prevent artificial passages of the discretized DNA strands at high compressive forces by using *k*_s_ = 80*ε*_0_, where *ε*_0_ = *k*_B_*T*. The second bonded interaction is used to introduce bending stiffness of the DNA molecules. The bending stiffness was modelled by harmonic interaction *U*_b_(*θ*) = *k*_b_(*θ* − *θ*_0_)^2^, where *k*_b_ represents penalty against bending of the DNA chain and *θ*_0_ is the equilibrium angle set to *θ*_0_ = π. The constant *k*_b_ is set to 20 *σ*/*ε*_0_ what corresponds to the experimentally common value of persistence length *P* = 50 nm [46,47]. The volume of the polymer chain is modelled by the non-bonded excluded volume interaction in the form of fully repulsive truncated and shifted Lennard-Jones potential *U*_ex_(*r*) = 4*ε*_0_ [(*σ*/*r*)^12^ − (*σ*/*r*)^6^ + 0.25], if *r* < 2^1/6^ *σ* and *U*_ex_(*r*) = 0 otherwise. The DNA in all simulations was considered nicked and torsionally relaxed. The torsional stiffness can be induced by using some of our previously developed models [48,49,50] for the price of increased computational costs. However, for small DNA chains below 1 kb the torsional relaxation of linear chains is very fast, and in our previous work we have also shown that the torsionally stiff molecule is able to quickly relax emerging supercoiling by effusion through chain ends, if the molecule is short enough (100 beads), so no differences on stereospecificity were observed as compared to simple model of DNA without torsional stiffness, (Figure S2 in [44]). Hence, our current model can represent DNA with 2 nicks 100 beads distant. All the simulations were performed by using Extensible Simulation Package for Research on Soft matter [51,52]. We performed Langevin molecular dynamics simulations with solving equations of motion *m*$\ddot{r}$ = −*γm*$\dot{r}$ − ∇*U*(*r*) + *R*(*t*)(2*ε*_0_*m*γ)^0.5^, where *r* represents position vectors of the beads, ∇*U* is the force acting on the bead calculated from bead’s interactions described above *γm*$\dot{r}=$
*ξv* is a damping constant in units of reciprocal time and the last term represents implicit random kicking force from surrounding media, where *R*(*t*) is a delta-correlated stationary Gaussian process. The equations of motion were integrated with the time step d *τ* = 0.01 *τ*. The physical dimension of the time unit [*τ*] = 74 ns [45]. After inserting polymer chain into the channel, we performed a pre-equilibration run of 10^9^ MD integration steps equal to 10^7^ *τ*, followed by the production run of the same duration, during which we collected 5000 samples for analyses, adding also 5 repeated production runs for each setting of confinement strength *D*/*P* and compression force *F* (see below). The length of our pre-equilibration and production runs is 10 times longer than used in the previous simulations works [35,36], but we did fewer repeated runs. We arranged our simulations in the way the compression uses longer “waiting time” instead of cycling compression/decompression cycles, based on the previous work that showed that an equilibrated entangled state was reached in one cycle when using long waiting times [36].

### 2.2. The Model of Helical Nanochannels

The confining nano-channels were modelled as a helical tube of constant diameter as measured in a cross-section perpendicular in the centre of the helical channel. The parametric equation of the mid-curve of the helical channel is given as *r*_0_ (*t*) = *k*$t\hat{i}$ + *R*_H_cos (ω*t*) $\hat{j}$ + *R*_H_sin (ω*t*) $\hat{k}$ [53], where *t* is a periodic parameter in radial space and ω gives a subtended angle as *t* increases and ω carries also information on the handedness of the helix whether it has plus, +ω, or minus, −ω, sign. Unlike cylinders, helical tubes are characterized by three additional parameters in addition to the diameter of the channel [53], *D* = 2 *R*_ch_: radius of the helix, *R*_H_, the pitch *k* denoting the distance between helical loops equal to *d*_H_ = 2 π*kσ* and handedness of the helical tube, sign (ω), that can be negative or positive. The radius of the helix in all simulations with helical channels is *R*_H_ = ⅓ *R*_ch_, based on our previous MD simulations that determined the strongest effect of channel helicity on chiral properties of knotted polymer for *R*_H_ in the range between ⅓*R*_ch_ to ½*R*_ch_ [44]. Additionally, simulations with *R*_H_ = 0 were performed in order to obtain trajectories for cylindrical geometry. Similarly, the setting of the pitch *k* = *D*/(2 π) was chosen based on our previous investigations of knotted polymers in helical channels [44]. We performed simulations with various ratios of confinement strength *D*/*P* = 2^i^ where *i* = −1, 0, 1, on the borders of strong, intermediate and weak confinement. In our previous MD simulations [44], we have used a helical tube that was modelled explicitly by constructing the walls of the channels using beads with diameter of 1 σ. Despite the positions of the explicit beads forming the walls of the channels being fixed during integrations (i.e., their equations of motions did not have to be solved), the pairwise excluded-volume interactions between the polymer and the walls still had to be computed, thus making the use of explicit walls computationally heavy. Because of the length of the intended simulation runs of 10^9^ integration steps per each trajectory, and especially also radius of the channel *R*_ch_ as free parameter, we developed an implicit model of the confining walls of helical channels. For this purpose, we needed to implement a function that computes the nearest distance of a bead to the helical wall provided along with its norm vector to be passed into the simulation engine of the simulation software ESPResSo. The developed implicit constraint uses a solution derived for solving mechanics of celestial bodies (Algorithm 1 in [54]), solving Kepler’s equation by Newton’s non-linear iterative method using an initial estimate for eccentric anomaly *E* (Equation (10) in [54]). The initial estimate made the MD simulations stable, converging and fast. Note, that in the case of a point laying outside of a helix, there could exist several valid solutions, however, we think this is not the case when a bead is contained inside a helical tube, 2 π*k* > *R*_ch_. We experienced that the implementation of the algorithm to model implicit helical confinement was fast and stable during hundreds of repeated runs with 3 × 10^11^ distance calculations performed for each bead in every trajectory. 

### 2.3. Piston Compression

In order to simulate the piston compression similar to the experimental setting of a gasket driven by mechanical tweezers, we adopted the approach used in the previous works [33,34,36] where we used a large bead that did not interact with the walls of confinement, but did interact with the polymer inside the channel, representing the piston. The confinement is modelled by the implicit model for cylindrical (*R*_H_ = 0*σ*) and helical (*R*_H_ = ⅓*R*_ch_) channels (see Section 2.2), also with an impenetrable wall at the bottom of the channels. The piston bead had a very large radius of 100*σ*. We used a range of compression forces covering region from weak compression forces *Fσ*/*ε*_0_ ≤ 0, 0.1, 0.35, 0.5 and 1 to the strong compression *Fσ*/*ε*_0_ ≥ 2, 5, 10, 15 and 20 similarly to previous works [34,35,36].

### 2.4. Topological Analyses

Topological analyses were performed by using specialized software KymoKnot [55] and Knoto-ID [56]. While KymoKnot is very fast in obtaining knot types, it implements a simple closure and Alexander polynomial invariant to identify knots. Knoto-ID can in addition find probabilistic knots and Jones polynomial invariant provides information on handedness of knots, however, solving the polynomial in dense/strongly compressed structures is computationally costly. Nonetheless, Knoto-ID can approach chirality also by computing Gauss diagrams that yield information on numbers of negative and positive crossings whose computation is much faster than computing knot types and provides direct information on handedness of the crossings of polymer’s entanglements.

## 3. Results and Discussion

### 3.1. DNA in Cylindrical versus Helical Confinement

Prior to going into results and discussion of compressed polymer chains, we focus on a simpler case when the DNA is subjected only to a single constraint imposed by confinement. Despite the case of cylindrical confinement being explored in a great extent previously by MC simulations [33,34], we have performed our own MD simulations of the DNA in cylindrical channels (with the setting of *R*_H_ = 0 in our implicit model for helical channels) in order to have access to all polymer metrics and properties that could show useful to demonstrate distinctive behaviour of the chain in channels with helical geometry. One of the most basic metrics of polymer chain is expressed by radius of gyration of the polymer and its components. 

In Figure 1a, we show the evolution of instantaneous values of gyration radius along the initial pre-equilibration molecular dynamics trajectory as obtained for DNA chains confined in helical and cylindrical channels with the confinement strength expressed as the ratios of the channel diameters to the chain’s persistence length indicated by numbers *D*/*P* = 0.5, 1.0 and 2.0. Along with the instantaneous values, running averages are also shown with solid lines. The employed colour scheme shows the values obtained for cylindrical channels in shades of black and the values obtained for channels with helical geometries are shown in shades of orange. The evolutions of the instantaneous values show the extent of fluctuations increases with increasing the diameter of the channel. This infers also insight on conformational transformations of DNA chains in the channels. First of all, in the case of the strongly confined DNA, the DNA chain is pre-stretched and remains mostly in an “I” conformation. As the channel widens, for *D*/*P* = 1.0 we observe emergence of backfolded conformations known as hairpins. The hairpins in the helical channels are only partially developed into one- and two-sided “J” shape conformations. In the case of the cylindrical channels, the hairpins develop a fully “U” shaped backfolded structures. Finally, as the channel opens, the DNA in weak confinement forms multiple folded structures indicated by using impression of letter “W”. The running averages of the gyration radii indicate that the DNA chains in helical channels evolve in slightly more elongated structures than in cylindrical channels with the same diameter. The evolutions of the instantaneous values of gyration radii together with running averages show also that the equilibration varies with the channel diameter. For the purpose of calculating average values with statistical standard error deviations we performed production runs of the same length in five repeated runs.

The observation of slightly larger gyration radii in helical channels is already surprising and provides a hint on how the helical geometry acts on the semi-flexible polymer chains. Helical channels have the same diameter as the cylindrical channels, but in addition, the helical channels do have a curvature that makes pathway, *C*, along the centre of the channels longer than in cylinders. The ratio of the length of the curved line, *C*, against the distance of two points, *L*, gives a parameter known as tortuosity, *τ*_t_ = *C*/*L*. Tortuosity increases the effective distance between two points on the helix, hence, if the polymer would localize mainly in the centre of the channels, following the channels curvature, its length would be shorter than the length of the polymer in cylindrical channel by factor of $f\;=\;{\left[{\left(2\;\pi k\right)}^{2}\;+\;R_{H}^{2}\right]}^{0.5}/2\;\pi k$
~20% [44]. Since the polymer in the helical channels appears to be slightly more extended than in the cylindrical confinement, it demonstrates that the interplay between the chain stiffness and the helicity of the channel does not allow the polymer to freely explore loops of the helix, and hence the helical confinement in terms of the polymer extension appears to be slightly stronger than the cylindrical one. In such case, the polymer would stay in the inner section of the channel that would have in our setting *R*_H_ = 0.3 *R*_ch_ diameter of *D*_in_ = 2/3 *D*. This is partially confirmed by the plot of the transversal component of the gyration radius, in Figure 1b. The transversal radius of gyration represents distribution of polymer into lateral sides of the channel [57]. Figure 1b shows, that the $R_{⊥}$ increases with the increasing diameter of the channel and decreasing the confinement strength *D*/*P* indicated on the plots by numbers *D*/*P* = 0.5, 1.0 and 2.0. At the same time, the running averages show that the distribution to the lateral sides of the channel is smaller for the helical channels. The average values of the lateral distribution by transversal gyration radius are not a sensitive quantity to fully capture the distribution of the chains across the cross section of the channels, and we will demonstrate this additionally later by calculating radial distributions of monomers, number densities on the surface and planar projections of the monomer distributions. The values of the transversal gyration radii however show that the difference between the values obtained in the helical channel and the cylinder differ less than by one third, hence the polymer does not reside exclusively in the inner section of the helical channel away from the helical grooves of the channel. 

Another common property to quantify effect of confinement is the extension of the polymer obtained in terms of the span of the chain. The span of the chain is defined as the maximum distance achievable between two monomers along the polymer chain, *R* = max (**r***_i_*, **r***_j_*), *i* ≠ *j* and *i*,*j*
∈ *N*. The span of the molecule can be obtained by using all cartesian coordinates *x*, *y*, *z*, sometimes denoted as *R*≡*S* (**r**) or, alternatively, by using only the coordinate along the major axis of inertia of the channel, denoted as *R*≡*S* (*x*). In our work, we are using primarily the span defined as *R*≡*S* (*x*), as the data on compression with large forces and very large diameters inferred a possible chain size bias when the chain laid flat on the bottom of the channel, and the values of the maximum span obtained as *S* (**r**) were distorted by showing non-monotonous behaviour with the compressive force. In Figure 1c, we show the extension of the DNA in terms of chain span *R* as a function of confinement strength *D*/*P*. The values are shown in black and orange colours for cylindrical and helical channels respectively, together with the statistical errors. Again, the pre-extension induced by helical and cylindrical channels results into very similar values, where the obtained difference is statistically reliable mainly in strong confinement *D*/*P* = 0.5. The values of the span obtained in the cylindrical channel is *R* = 266.7 ± 3.3 *σ* and for the helical channels *R* = 271.2 ± 2.6 *σ*, hence the difference between the values of the *R* is larger than is the calculated standard deviation error. The values in the region between *D*/*P* = 0.5 to 1.0 can be well approximated by theoretical prediction given by equation *R* = *L* [1 − *A* (*D*/*P*)^2/3^] (the dashed line in Figure 1c, where *A* = 0.1701 was obtained for a cylindrical channel [58]. The value of *A* seems to be slightly smaller for the helical geometry, but since we have only a single data point and the difference from predicted value is within the statistical error, we currently use this value as satisfactory for prediction of the polymer span also for the helical channels (with *R*_H_ < *R*_ch_). In the region between *D*/*P* = 1.0 to 2.0, the dependence of the chain extension is approximated by a line corresponding to a power law fit that predicted the exponent *R* ≈ (*D*/*P*)^−1.02^ (the dotted line, in Figure 1c [17]). The small differences arising from helical and cylindrical geometries are probably related to the setting of the pitch of the helical channels. The pitch of the helical channels, *k*, is set so that the distance between helical loops, *d*_H_, scales with the strength of confinement *d*_H_ = ½*P*, *P* and 2 *P* respectively. The deflection length of a polymer in cylindrical channel of diameter *D* is *λ* = *D*^2/3^ *P*^1/3^ [16], that corresponds to 12.6, 20 and 31.7 *σ* in the investigated confinement strengths *D*/*P* = 0.5, 1.0 and 2.0. Since the deflection length is slightly larger *λ* > *d*_H_ only for the regime of *D*/*P* = 0.5, this might be responsible for observing significant differences in the scaling of the polymer metrics between helical and cylindrical channels only in the strong confinement regime.

In Figure 2, we show probabilistic distributions of the chain span that is also often used to evaluate the effect of confinement on polymer chains. The distributions were obtained by counting frequencies of occurrence of chains with a given value of the span, *R*, and the bin size of the histograms set to 1 *σ* from trajectories obtained during five repeated productions runs, each consisting of 5000 frames for analyses. The plot shows comparisons of probability distributions obtained for different confinement strengths *D*/*P* = 0.5, 1.0 and 2.0 and channels with cylindrical and helical geometries (black and orange lines). The main difference of probability distributions obtained in helical and cylindrical channels is observed for confinement strengths *D*/*P* = 0.5 and 1.0, where the peaks in helical channels are notably positioned toward higher values. The peaks of the probability distributions calculated for helical channels are also narrower. At weaker confinements, the distributions show also asymmetries with extension of left or right tail of the distribution that was reported in the earlier works. The probability distributions of the chain span are related to the elastic free energy *A* (*R*) = *c* (*T*) − *k*_B_*T*lnP (*R*), while by differentiation one obtains force *F* = −d*A* (*R*)/d*R* that acts on the endpoints of the chain in attempt to restore unperturbed equilibrium properties. It has been shown, that this method originally developed to study elasticity of polymers, can be used to calculate the pre-stretching force applied by the confinement to the polymer chain [32]. The computed probability distributions indicate existence of a higher pre-stretching force in helical channels that will counteract also the external compressive forces. As an inlay in Figure 2, we show also representative snapshots of prevalent conformations of the polymer chain together with simplified impressions given by letters, as discussed above along with the discussion of Figure 1a.

In addition to the transversal gyration radii, we have calculated radial distributions of monomers along the cross-section of the channels. These were obtained by counting monomers in concentric shells in a binned distance from the centre of the channel. The bin size was set to 0.5σ, hence the volume of the layer corresponded to 2 π*L*_ch_ (0.5 σ)^2^ [(*n* + 1)^2^ − *n*^2^]. The distributions were normalized in the way that the obtained curves represent the probability of finding a monomer of the chain with length *N* in the radial distance from the centre of the channel, *r*/*R*_ch_. It is also important to note, that in the case of helical channels, in order to calculate the distance from the helical centre, we employed the same algorithm described in [54] that was implemented to calculate the distance of beads from walls during solving equations of motions in molecular dynamics simulations (see Section 2.2). The data in Figure 3a show that the radial distributions of monomers are mis-shaped as compared to channels with cylindrical geometry as a result of tri-axial symmetry breaking in the channels with helical geometry. The distributions in the helical channels shows slight increase of the monomer concentration in the middle of the channel (*r*/*R*_ch_ = 0). Then, the line crosses the distribution computed for cylindrical channels two times, suggesting that the monomers would concentrate on the surface of the inner ridges (threads) of the helical grooves. In addition, we have calculated an integral of the number density of monomers ($\varphi_{0}$) at the surface layer *δ* = ⅕ *σ* thick (inset of Figure 3a). By this approach, that we used also earlier on studies of entropic segregations, we have calculated the confinement free energy *A*_C_ $=2\pi L_{\mathrm{ch}}\int \varphi_{0}\left(r\right)dr$_,_ for *r* ∈ <*R*_ch_ − *δ*; *R*_ch_ > [29]. The calculation shows that despite smaller lateral distributions of monomers, indicated by transversal gyration radii and radial distributions also indicating increase of the monomer concentration in the middle of the helical channels, the distribution of the monomers in the cross section of the channel concentrates around inner ridges of the helical grooves (see also heatmaps in Section 3.2). The increased concentration of monomers in the inner channel with ⅔*D* diameter is also demonstrated by computing radial distribution function from the major axis of inertia of the channel (Figure 3b).

The effect of confinement and the situation of the chain within confining channels is also often explored by calculation of bond orientational correlations functions. The orientational correlations functions computed in channels with three different diameters represented by the confinement strength *D*/*P* = 0.5, 1.0 and 2.0 are shown in Figure 4, for channels with helical and cylindrical geometries distinguished by orange and black. In general, the orientational correlation functions show three distinct regions [59]. First of all, on short scales with *s* ≤ *D*, the curves show an onset of the exponential decay typically encountered in unperturbed conditions of a polymer chain in the bulk. The onset in confinement is characterized by shallow minima beyond which the effects of confinement start to be visible. In the region of *s* > *D*, the correlations behave differently based on the confinement strength. In the strong confinement characterized by *D*/*P* = 0.5, the orientational correlations develop a broad plateau where the interplay of confinement and the chain persistence length create an apparent stiffening of the chain into a rod-like structure [60]. The computed values indicate that the apparent stiffening is larger for helical channels as the monomers hit the inner ridges of the helical grooves. For the channel with intermediate strength of confinement *D*/*P* = 1.0, the orientational correlations computed for helical and cylindrical channels exhibit the largest differences, as the helical geometry of the channel probably extends the Odijk’s regime towards the transition region. In the case of the channels with the largest diameter investigated here, with *D*/*P* = 2.0, the orientational correlations in cylindrical channels drop below the values in the bulk indicated by dashed line in Figure 4, the effect known as polymer softening [32,33]. The simulations in the helical channels reveal that the softening effect is not observed. Finally, the third region on the graph of orientational correlation functions can be identified by *s* → *L*, or the very tail of the functions. This is where the wiggling of the polymer tails comes into action to quickly nullify the extent of orientational correlations.

### 3.2. DNA under Compression in Cylindrical and Helical Confinement

After investigating the effects of confinement on DNA chain in channels with cylindrical and helical geometry, we gained insights to explore also the situation when the chains are subjected to the external force applied. We expect that the compression in terms of the chain extension will be smaller in the helical channels, as the chains in the helical confinement exert more extensive force to the chain ends due to the larger elastic free energy. Figure 5a,b show the chain extension as a function of the external compressive force. The plot is divided and the extension is shown for weak and strong compressive forces separately in order to enhance readability of the displayed values. The values for cylindrical channels are shown in black and the values obtained during the simulations in helical channels are now shown in orange and blue lines, also distinguishing handedness of the channels (blue for negative, and orange for positively wound channels in all figures). Figure 5a infers that major differences between polymer spans *R* obtained for different compressive forces are observed for strong confinement, *D*/*P* = 0.5 and weak compressive forces. Here, the dependence of *R* is non-monotonous, starting with a plateau region that was observed by earlier MC simulations of linear and pronounced in ring polymers [34]. Figure 5a shows, the plateau region obtained for linear polymers in helical channels is slightly extended towards *Fσ*/*ε*_0_ = 0.35, while it vanishes at *Fσ*/*ε*_0_ = 0.2 in cylindrical channels.

The strong confinement enhances apparent elastic persistence length of the DNA chains, so they effectively behave like rods stretched by confinement into “I” conformation (snapshots in Figure 2 and Figure 6a). Consequently, with the interplay of weak compressive forces, the chain may give rise to the effect similar to Euler’s buckling [35]. When higher loads of compressive forces are applied *Fσ*/*ε*_0_ = 0.5, the chain already forms double-backfolded “U” shaped hairpins and partially triple-folded structures (snapshots in Figure 2 and Figure 6). For the compressive forces *Fσ*/*ε*_0_ = 0.5 we did not observe higher order folding in the strong confinement *D*/*P* = 0.5. For strong compressive forces above *Fσ*/*ε*_0_ > 1, data computed in Figure 5a indicate collapse of the chain span, which is illustrated by the snapshots in Figure 6a. At very high compressive forces, *Fσ*/*ε*_0_= 20, the chain exhibits effect known as spooling encountered in DNA tightly packed in bacteriophages [62,63] and viral capsids [64]. The orientation of the spools in the narrow channels, *D*/*P* = 0.5, is longitudinal with the chain aligning with the direction of the main axis of the inertia of the channel while their orientation changes in larger channels and they wing around the main axis of the inertia of the channel (Figure 6a–c). In the helical channels, the spools are also distorted and skewed following the helical curvature of the channel. The monomer radial distribution function for helical and cylindrical channels obtained for *D*/*P* = 0.5 are compared in Figure 6d for 3 compressive forces *Fσ*/*ε*_0_ = 0.1, 1 and 20. The comparison shows that with increasing compressive force the monomers shift towards the walls, while the monomer concentration in the middle of the channel decreases. This process is more prominent in cylindrical channels where in the case of the highest compressive force *Fσ*/*ε*_0_ = 20 the monomers are expelled from the middle of the channel with the probability of finding the monomers close to the walls being higher than in the middle of the channel.

The plateau region is not observed for chain span in intermediate and weak confinement *D*/*P* = 1 and 2. In general, the computed extension-force curves show a continuous behaviour with decrease of the polymer span on increasing compressive force. With increasing the compressive force, the differences between the polymer extensions computed for helical and cylindrical channels disappear within the significance given by the standard error deviations. In Figure 5c, we show the computed dependencies of the chain span versus the compressive force in log-log representation. In this representation, the computed curves follow the established linear behaviour *R* ≈ *F*^Y^ with the exponent *Y* = −9/4 = −2.25 [38], shown by the dashed line. The concatenated fits over our computed data yielded a value of the exponent *Y* = 2.108. The deviation from the established value arose mainly from the plateau region at *D*/*P* = 0.5 that is not in the interval of values of *D*/*P* where the theoretical fit was originally designated.

The 3D molecular snapshots from the MD simulations in Figure 6a–c show that as the diameter of the channel increases, weaker forces come in place to collapse the polymer chain. As a difference to the case of *D*/*P* = 0.5, at very high compressive forces and large channels *D*/*P* ≥ 1, the polymer is found laid flat on the bottom of the channels. The polymer spools are now winding around the main axis of inertia of the channel aligning with the radial coordinate of the channel, with monomers concentrating around the walls of the confining channel and away from the centre as observed in previous work [39]. This is revealed by the snapshots from the molecular simulations, shown in Figure 6a–c. A very pronounced effect of polymer spooling is demonstrated also by the radial distribution functions, Figure 6d, that show the probability of finding a monomer in the middle of the channels with *D*/*P* ≥ 1 drops almost to zero for very high compressive forces.

Along with the 3D snapshots of the molecular structure in the channels, the planar projections of radial distributions of monomers in the channels are shown as heat maps, similar to Figure 2 in [65]. The snapshots and the heat maps also demonstrate that at very high forces *Fσ*/*ε*_0_ ≥ 5 and wider channels characterized by *D*/*P* = 1 and 2, the DNA chains are very much collapsed, so that the molecular span *R* is smaller than the radius of the channel. In relation with this observation, we expect a chain length bias can be encountered in the data. For example, in Figure 7, we show the number concentration of monomers on the surface of the channels computed as number of monomers in a thin layer of *δ* = ⅕ *σ* that corresponds to the confinement free energy [29]. We observe, that for strong confinement *D*/*P* = 0.5, where the *R* of the molecule does not drop below the diameter of the channel, the number concentration of the monomers obtained in helical channels is larger as compared to the cylindrical ones in the whole investigated range of the compressive forces, consistently with the observations made for uncompressed polymers discussed in the previous Section 3.1. On the other hand, in the case of larger channels with *D*/*P* ≥ 1, the concentration of monomers obtained for helical channels crosses the curve obtained in cylindrical channels (indicated by arrows). The computed concentration dependences cross at *Fσ*/*ε*_0_ = 2 for *D*/*P* = 2 and at *Fσ*/*ε*_0_ = 1 for channels with *D*/*P* = 2. The chain size bias is also related to the fact that the concentration of monomers on the bottom is not counted to evaluate effect of walls, and the cross section of helical channels has larger area than the cylindrical geometry.

### 3.3. Topology of DNA under Compression in Helical and Cylindrical Confinement

After investigating and comparing effects of confinement and compressive force on basic metrics of DNA chains in nanochannels with helical and cylindrical geometries, we further proceed by investigating changes in polymer topology. As we discussed earlier, geometry of the polymer chain and geometry of confining channels coexist in a two-way relationship, that in turn affects the resulting topology of the semiflexible chain. In strong confinement, the polymer exhibits an enhanced orientational persistence length giving rise to rod like “I” conformations. These would further collapse under compressive forces in multiple folded structures, like incomplete “J” and fully developed hairpins “U”, or even more complex structures like polymer spools. The topological changes are often investigated by orientational correlations, but, in the case of biological molecules such as proteins and DNA, the topology is often evaluated in terms of knot theory.

The orientational correlations computed for compressed DNA chain confined in the cylindrical and helical channels with different confinement strengths *D*/*P* = 0.5, 1.0 and 2 are shown in Figure 8a–f, respectively. The plots show five lines corresponding to different compressive forces, *Fσ*/*ε*_0_ = 0.1, 0.5, 1.0, 5.0 and 20.0, and a dashed line showing the exponential decay of the orientational correlations in the bulk for reference, obtained as <cos *θ*> = exp (−*s*/*P*) [61]. The orientational correlations computed for very small compressive force *Fσ*/*ε*_0_ = 0.1 show very similar behaviour as observed for uncompressed confined DNA, shown in Figure 4 and discussed in Section 3.1. The orientational correlations show three distinct regions given by separation of monomers along the chain *s*/*σ*. First, at *s* ≤ *D*, the correlations show an onset of exponential decay delimited by a shallow minimum, next evolving a plateau region in strong confinement and producing elastic stiffening at *D*/*P* = 2, and finally at *s* → *L* the orientational correlations quickly drop as a result of random motions of polymer ends.

For stronger compressive forces, the compressive force competes with elastic free energy (see discussion to Figure 2, in Section 3.1). This in turn shifts the shape of the orientational correlations in the mid-region delimited by *D* < *s* < *L* towards behaviour observed for weaker confinements. As the compressive force increases, double folded “U”-shaped hairpin structures in strong cylindrical confinement (*D*/*P* = 0.5) are represented by “V” shaped orientational correlation functions with anticorrelated negative values often encountered in ring polymers [34,64].

At very high compressive forces, the orientational correlation functions exhibit oscillatory behaviour while the computed lines at first drop below the dashed line (representing the decay of unperturbed DNA), hence indicating elastic softening under strong compression. This oscillatory behaviour of the orientational correlations was associated with formation of spooling at densely packed polymers [64]. The oscillatory behaviour of the orientational correlations is not observed for narrow helical channels. The reason is that the spooled polymer chain is also helically skewed and twisted around the main axis of the channel (see snapshots in Figure 6a). In larger channels, *D* ≥ *P*, the frequency of oscillations drops with increasing diameter of the channels that determines also the diameter of the spooled structures and increases arc length and separations between polymer turns.

Next, we investigated topology of the confined compressed chains in terms of knot theory. This type of analysis provides information on degree of knotting and identifies knot types created by polymer self-entanglements. Thus, this analysis is particularly appropriate when studying topological changes induced by knot factories as it probes the property that is manipulated in these devices. For the purpose of the analysis, we used two software packages, KymoKnot [55] and Knoto-ID [56] that perform knot search on discretized curves, such as beaded polymer chains from molecular simulations, whose 3D coordinates are loaded as an input at the beginning of the analysis. Both of the computational tools adopt three essential elements during the knot search. First of all, since the knots mathematically do exist only on closed curves a method for constructing a closure between two ends of a (sub)chain needs to be chosen. This step is vulnerable for a potential bias induced by the closure method chosen. The KymoKnot implements minimally interfering closure scheme that minimizes probability of inducing new entanglements by the closure method [66]. In the case of Knoto-ID, the computational tool allows choosing from various approaches to the closure by making projections and computing also probabilistic knots and knotoids by computing the overall frequencies of occurrence. Another essential element is a simplification of the curve that removes the crossings without changing topological state of the curve. This step may also involve pre-smoothing of the curve, as implemented in the KymoKnot, that in turn speeds up the consequent simplification of the curve by Reidemeister moves [67,68]. Finally, a chosen polynomial invariant is evaluated for the obtained knot diagram. KymoKnot uses Alexander polynomial evaluated in *t* = −1 and Knoto-ID uses Jones’s polynomial as topological invariants. The Alexander polynomial is the fastest algorithm available [69], which becomes especially crucial for analyses of knotting in densely packed polymers, such as we obtained in strong compression modes, where computational time needed for calculations rapidly increases. A drawback for using the Alexander polynomial is that it is a simple invariant that provides only limited properties of knots and cannot distinguish knots based on chirality. Jones polynomial is able also to distinguish chirality of knots [70]; however, the algorithm takes a lot of computational time beyond feasibility of conducting more extensive calculations for the very dense/compressed polymer chains. For the purpose of analysing also chirality of knots, we used calculation of Gauss linking number implemented by extended Gauss code in Knoto-ID software [56,71] (see below).

In Figure 9a, we show knotting probabilities obtained for the DNA chain under compression and confined within helical and cylindrical channels with three different diameters characterizing confinement strength (*D*/*P*). The probabilities are shown as stacked areas representing probability of occurrence of certain type of knot, characterized by its number of crossings, *k*. The knotting probabilities corresponding to different channel geometries are arranged in columns, indicated by labels −ω and +ω for helical channels with negative and positive handedness, respectively. The data for different confinement strengths *D*/*P* are arranged in rows with the particular value of *D*/*P* shown as numbers 0.5, 1.0 and 2.0. The computed knotting probabilities are shown in a colourmap scale, starting with plain blue colour representing unknots, and ultimate plain red colour representing complex knots with the number of crossings larger than *k* > 11. In-between, knots starting from two chiral forms of trefoil 3_1_ and 3_1_ m, and single achiral knot 4_1_, are shown, to more complex knots that could be identified and named according to the Rolfsen table [72]. These are shown with different shades of colour and separated by lines, together with the colour scale of *k*_n_ in the associated legend, where the number indicates the number of crossings and index *n* denotes the knot type as shown in the Rolfsen table.

The first row on the composite graph in Figure 9a shows the knotting probabilities obtained for very strong confinement with *D*/*P* = 0.5 in channels with right-handed and left-handed helical geometry and compared to channels with cylindrical geometry, as a function of increasing compressive force. The direction of the increase of the force is indicated by an arrow on the bottom of the columns. The graphs show, that at very small compressive forces *Fσ*/*ε*_0_~0.1 no knots are detected, what is consistent with observation of extended “I” shaped conformation of chains, as discussed with Figure 1 in Section 3.1 and as shown also as snapshots in Figure 6a, and indicated also by orientational correlations functions in Figure 8a,d. As the compressive force increases *Fσ*/ε_0_~1, the chain becomes more readily folded with emergence of simpler knots with smaller number of crossings indicated by the computed knotting probabilities. At even higher compressive forces 1< *Fσ*/*ε*_0_ < 5, one can observe that very complex knots with crossing number larger than 11 come more and more into play to represent topology of the chain. The probability of occurrence for the spectrum of simpler knots, with crossing number between 3 to 11, also widens. At very high compressive forces, *Fσ*/*ε*_0_ > 5, the unknots (0_1_) are heavily suppressed, and the knotting probability is dominated by the occurrence of very complex knots. At the same time, also the probabilities of the spectrum of simpler knots are suppressed. The distribution of knotting probabilities in the cylindrical channels appears to be, however, much wider that that observed for helical channels, suggesting that the helical geometry of the channels enhances knotting as compared to cylindrical channel.

In the case of knotting probabilities in weaker confinement *D*/*P* = 1 (the middle row in Figure 9a), a similar behaviour of knotting probabilities to the one described for strong confinement can be observed. The dependence of the knotting probability on the compressive force in right-handed and left-handed helical channels is very similar, with monotonous drop of probabilities of unknot and simpler knots, while the spectrum of intermediate knots (with crossing numbers between 3 to 11) is also much wider compared to that computed in strong confinement. For small compressive forces, *Fσ*/*ε*_0_ < 1, the probability of them existing in unknotted state is however smaller, than observed for strong confinement. This is can be related to the fact, that at weaker confinement smaller compressive forces are needed to induce folding, as shown in Figure 6b and discussed also with the orientational correlations and effect of elastic softening in Figure 8. At high compressive forces, *Fσ*/*ε*_0_ > 5, the computed data start showing instability prominent especially for cylindrical channels. As we discussed earlier, this instability was foreseen as we became aware of the size bias of the chain due to the finite number of monomers and rapid increase of volume in wider channels explored (discussion to Figure 7). As the conformation of the polymer transforms to the spooled structure, the knotting probability associated with this transformation may drop. In the case of the helical channels, they still induce some asymmetry due to tri-axial symmetry breaking in the helical geometry, hence, the monotonous appearance of the knotting probability dependence is not so affected.

In the case of weak confinement with *D*/*P* = 2 (bottom row of Figure 9a), the knotting probabilities start with prevailing unknot probabilities, that quickly drop even for very small compressive forces *Fσ*/*ε*_0_ = 0.1. This is consistent with observation of spontaneous knot formation on DNA confined in nano-channels with diameter 100 nm [73,74]. At higher compressive forces, *Fσ*/*ε*_0_ ≥ 1, the knotting probabilities seem to stabilize to a plateau region, as the chains are already spooled at the bottom of the channel, as indicated by Figure 8c,f. The knotting probabilities in helical channels still exhibit formation of complex knots, with a crossing number larger than 11, that is not observed for cylindrical channels, probably as the helical channels still maintain some effect of asymmetry acting on the chain.

In Figure 9b, a summary for knotting probabilities obtained for complex knots with crossing number *k* > 11, and confinement strengths *D*/*P* = 0.5, 1.0 and 2.0 is shown as a function of compressive force and compared for channels with helical and cylindrical geometry. The data computed for helical channels are averaged and shown with orange colour, and the knotting probabilities of complex knots computed for cylindrical channels are shown in black, together with the standard error deviations indicated by error-bars. The filled area corresponds to the difference between the probabilities obtained in helical and cylindrical channels. The comparison infers that the knotting probabilities in helical channels in general exceeds the probabilities of knotting in cylindrical channels, and the helical geometry of the channels enhances knotting.

In Figure 9c, we show the minimal average crossing number (*ACN*) that is another property used in knot theory to evaluate the topology [75]. The *ACN*, as a measure of topological entanglements, has been shown to be mildly dependent on the noise associated with the experimental setting but strictly dependent on the compressive force [36]. In order to obtain the *ACN*, we used Knoto-ID software that does not calculate *ACN* by default, but prints number of crossings of initial structure, then after smoothing by the 3D triangle elimination [76], and after simplification by Reidemeister moves as screen output. The *ACN* uses 3D curve and does not employ closure of the curve. The *ACN*’s were computed as averages of random 2000 projections for each chain (frame). The computed values show miniscule differences as obtained for right-handed and left-handed helical channels but they show significant differences between helical and cylindrical channels. The differences are significant as compared to the standard error deviation even at the high compressive forces. We speculate, that the enhancement of knotting observed by knotting probability and *ACN* arises from the helical grooves that may act as irregularities on surface of the channels, that may enhance knotting as pointed out by earlier work [41], rather than from a change of the volume. In fact, since the diameter of the helical and cylindrical channels in cross-section is the same, while the span of molecule in cylindrical and helical channels is very similar, the chain in helical channels with radius a *R*_H_ has more space available around it.

So far, we have shown by calculations of topological properties in terms of knot theory, i.e., by the computed knotting probabilities and average crossing numbers, but also complemented by radial distributions functions and raw projections of polymer distributions and snapshots, that the helical geometry of the channels enhances knottiness. However, while some of the evaluated properties show more pronounced differences between cylindrical and helical channels than others, the polymer extension, orientational correlations, knotting probabilities, average crossing numbers, projections, etc. are perfectly symmetric for right-handed and left-handed channels. As the aforementioned properties are related to the energy (as discussed along with Figure 2, Figure 3a and Figure 7), they are not apt to demonstrate any symmetry breaking. Stereospecific differences have to be revealed by topological descriptors that carry also information on the chirality. Such the descriptor is writhe of the chain given by Gauss integral [77]. As reviewed in the Introduction, several works reported emergence of helical structures of polymers under the compressive force [31,35,37,39,40]. Such helical structures would translate into writhe whose absolute value yields information on how many times the chain turns around itself, but the absolute value also comes with an associated sign that corresponds to the overall handedness. The overall writhe, however, is given by an average number of crossings with positive and negative signs, i.e., it represents an excess value of the two, which in addition do not necessarily originate in all extent from knottedness. This is similar to the situation of supercoiled knots, where the writhe of the knot does not correspond to its equilibrium value, but it carries also contribution of the chain turns originating from the supercoiling [78]. In our case, the chains are torsionally relaxed, but the additional winding can be imposed to the chain by the helical geometry of the channels.

In Figure 9a, we have shown and discussed knotting probability of identified types of knots. Some of these knots are composed from under-crossings and overcrossings, that following the direction of the chain can be identified as crossings with positive or negative sign, that will cancel out in the summation of writhe. In Figure 9c, the average crossing numbers do not distinguish between positive and negative crossings. Hence, in order to investigate stereospecificity of the compressed chains, we first computed writhe of the chain. For computation of the writhe of the chain we used our own libraries [79] that implement discretized solution for Gauss double integral [80]. Then we used the Gauss code implementation, that after smoothing by 3D triangle elimination and simplification by Reidemeister moves gives a Gauss diagrammatic representation of knots that contains information on number of positive and negative crossings and thus the overall writhe of a knot (knotted core) [56]. Figure 9d–f show the result of the analysis, where the lines show the overall writhe of the chain in blue and orange colours based on the obtained handedness. The calculations were made for left-handed (negatively wound, −ω), and right-handed (positively wound, +ω) [81] helical channels. By using the Gauss code implementation in Knoto-ID, we have analysed positive and negative crossings of the knotted core. These were counted and plotted in Figure 9d–f as areas. The graphs show that the overall writhe of the chain is significantly determined by the handedness of the helical channels. The knotting analysis furthermore indicates that the portion of the writhe that comes from the topological knots, and that chirality of the knots contributing to the overall writhe, in terms of the writhe of the knotted portion is also imposed by the handedness of the channels. The stereospecific effect of the helical channels evaluated in terms of writhe of the chain and knotted core shows, that the effect is stronger for smaller compressive forces in channels with *D* = 1 than in channels with *D* = 0.5, as lower compressive forces are needed to overcome elastic persistence length of DNA in wider channels. However, the stereospecific effect exhibits maxima, that is most likely related to the chain size bias. Regarding Figure 9e,f, it is also important to emphasize, that at the compressive forces above the value where the chain size bias starts taking place, the stereoselectivity of the channels vanishes and the writhe for different channels just takes values without any trends about x-axis. In the case of the channels with *D* = *P*, we show values in the force range *Fσ*/*ε*_0_ = <0;5>, and in the case of the channels with *D* = 2 *P*, the data unaffected by chain bias reach only to *Fσ*/*ε*_0_ = 1.

## 4. Conclusions

By means of molecular dynamics simulations, we studied the compression of DNA in helical channels defined by diameter of the channel, *D* = 2 *R*_ch_, radius of the helix, *R*_H_, pitch of the channels, k, and their handedness distinguished by the sign. The investigated were channels with three different widths, in order to investigate also the effect of the confinement strength, *D*/*P* = 0.5, 1.0 and 2.0. The radius of the helix was chosen based on our previous work that showed the chiral properties were exhibited in maximum extent for *R*_H_ = 0.3 *R*_ch_. And stereospecific effect vanished as *R*_H_ → *R*_ch_. The pitch of the helical channels was chosen *D*/(2 π) in order to maintain a geometrical self-similarity of the channels upon scaling dimensions of the channels.

The investigation of the confinement effect of the helical channels without compressive force present revealed that monomers of the DNA chain tends to localize closer to the major axis of the inertia and they do seldom fully explore helical grooves of the channels. As a result, the helical channels act as cylindrical channels with a slightly decreased diameter. The distance between helical loops of the channels, *d*_H_, scales with the confinement strengths 10, 20 and 40 *σ*. In these conditions, the deflection length in the channels is similar to the distance of the helical loops *λ* = *D*^2/3^ *P*^1/3^ = 12.6, 20 and 31.7. This is consistent with the main scaling differences between helical and cylindrical channels observed in the channels with *D*/*P* = 0.5.

When compressive force is applied to the DNA confined in the channels, we report the compression curves follow the predicted behavior with exception of the plateau region corresponding to strong confinement with weak compressive forces where the original scaling behavior was not derived.

While the common polymer metrics show only miniscule differences as obtained for DNA polymer under compression in helical and cylindrical channels, the striking differences are observed on topological properties explored by orientational correlations functions, knottiness, and writhe of the chain, but also chirality of the structures in terms of the sign of the writhe and handedness of the knots. The topological analyses in terms of the knot theory show that knot factories with helical geometry enhance knotting and induce handedness to knots in terms of abundance of crossings with the sign equal to handedness of the channels.

## Figures and Tables

**Figure 1 polymers-14-04201-f001:**
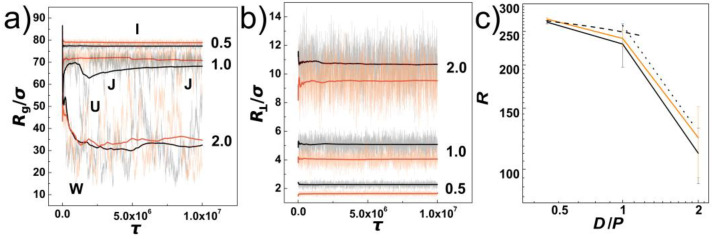
DNA confined in nano-channels without external compressive force. Data for cylindrical channels are shown in black and helical channels are in orange. The confinement strengths investigated are indicated by numbers *D*/*P* = 0.5, 1.0 and 2.0. Panels (**a**,**b**) show evolution of instantaneous values of gyration radius and transversal radius from the initial structure. The letters I, J, U, and W show impressions of DNA conformations. (**c**) average values of polymer span as a function of confinement strength *D*/*P*. The dashed and dotted lines show theoretically predicted values.

**Figure 2 polymers-14-04201-f002:**
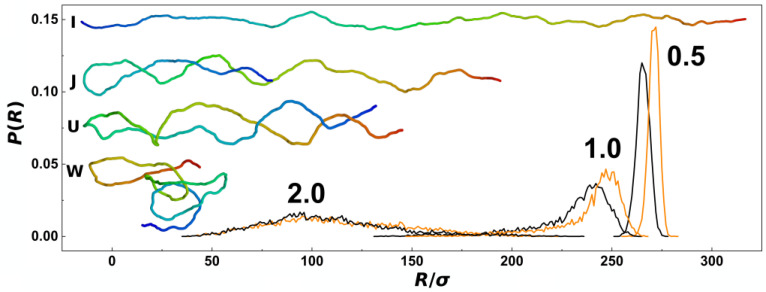
Histograms showing probability distributions of DNA span, *R*, obtained from simulations of DNA confined in nano-channels without external compressive force. The histograms obtained for cylindrical and helical nano-channels are shown in black and orange colours respectively. The panel shows also snapshots of the DNA chain in a rainbow colour scale from simulations for typical conformations represented by simplified impressions using the letters I, J, U and W. The numbers indicate confinement strength indicated as the *D*/*P* ratio 0.5, 1.0 and 2.0.

**Figure 3 polymers-14-04201-f003:**
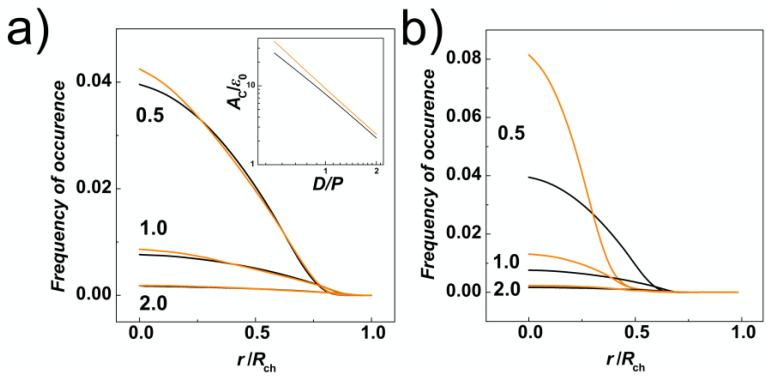
Radial distributions of DNA monomers obtained from simulations of DNA confined in nano-channels without external compressive force and calculated (**a**) from the centre of the nano-channel and (**b**) from the major axis of inertia of the channels in normalized radial coordinates, *r*/*R*_ch_. The values computed for cylindrical and helical geometries of the channels are shown in black and orange colours respectively. The numbers indicate regime of the confinement strength expressed as the ratio *D*/*P*. The inlay on the panel (**a**) shows also the dependence of the confinement free energy, *A*_C_, obtained as the integral of the monomer concentration on the surface of the channels [29].

**Figure 4 polymers-14-04201-f004:**
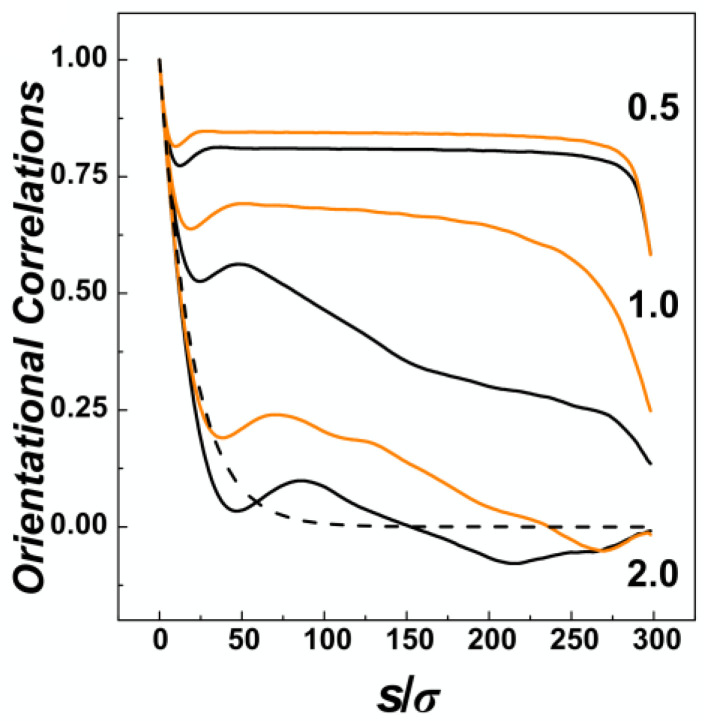
The orientational correlation functions along the coarse-grained curvature *s*/*σ* of the DNA polymer as obtained from simulations of DNA confined in nano-channels without external compressive force. The values obtained for cylindrical and helical channels are distinguished by black and orange colours and the strength of confinement in terms of the ratio D/P is indicated by number 0.5, 1.0 and 2.0. The the dashed line corresponds to the decay of orientational correlations of unperturbed DNA in the bulk <cos*θ*> = exp (−*s*/*P*) [61].

**Figure 5 polymers-14-04201-f005:**
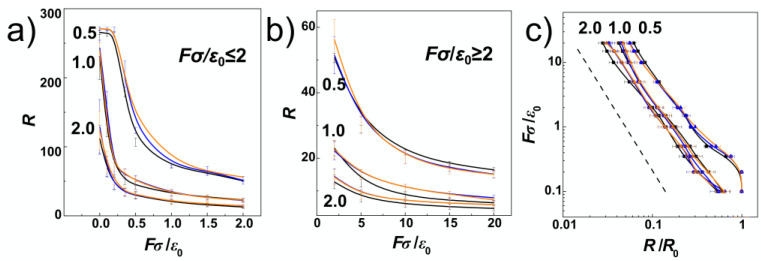
The evolution of polymer metrics during compression of DNA in helical and cylindrical nano-channels. In the plots, the black lines correspond to cylindrical channels, blue lines and orange lines to helical channels with negative −ω and positive +ω handedness respectively. The regime in terms of confinement strength expressed as the ration D/P = 0.5, 1.0 and 2.0 is indicated by numbers along the computed values of the span. Panels (**a**,**b**) show average extension of the DNA, *R*, for weak *Fσ*/*ε*_0_ ≤ 2 and strong *Fσ*/*ε*_0_ ≥ 2 compressive forces respectively. (**c**) Plot shows the extension of the DNA in log-log scale, while the dashed line corresponds to the theoretically predicted relation *R* ≈ *F^Y^*, where *Y* = −9/4.

**Figure 6 polymers-14-04201-f006:**
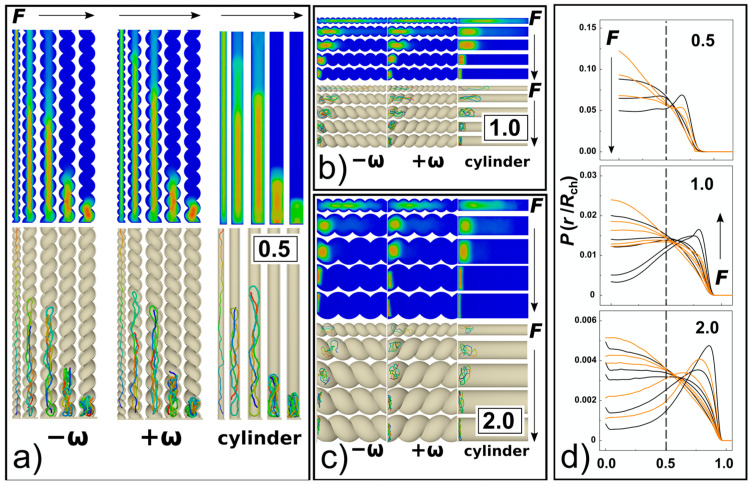
The compression of DNA in helical and cylindrical nano-channels. Panels (**a**–**c**) show planar projection heatmaps of distributions of DNA monomers across the channel together with snapshots for cylindrical channels and helical channels with negative and positive handedness. *F*’s with the arrows indicate the direction of increasing force (*Fσ*/*ε*_0_ = 0.1, 0.5, 1, 5, 20); (**d**) the panel shows the radial distribution function of the DNA monomers along the radial coordinate of the channel, for *D*/*P* = 0.5 shown with three lines *Fσ*/*ε*_0_ = 0.1, 1 and 20; and *Fσ*/*ε*_0_ = 0.1, 0.5, 1, 5, 20 for *D*/*P* ≥ 1.

**Figure 7 polymers-14-04201-f007:**
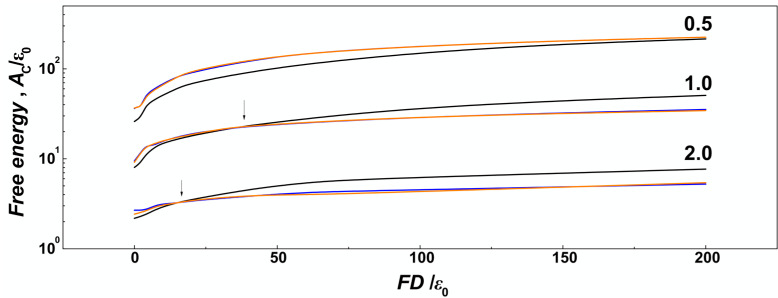
The confinement free energy during compression of DNA in helical and cylindrical nano-channels. In the plots, the black lines correspond to cylindrical channels, blue lines and orange lines to helical channels with negative −ω and positive +ω handedness respectively. Graph shows confinement free energy *A*_C_, obtained as the integral of the number density of monomers on the surface of the channel in a layer *δ* = ⅕ *σ* thick. The regime of the confinement strength is indicated by the numbers along the lines in terms of the ratio *D*/*P* = 0.5, 1.0 and 2.0. The arrows indicate where the chain size bias is expected to take place in larger channels.

**Figure 8 polymers-14-04201-f008:**
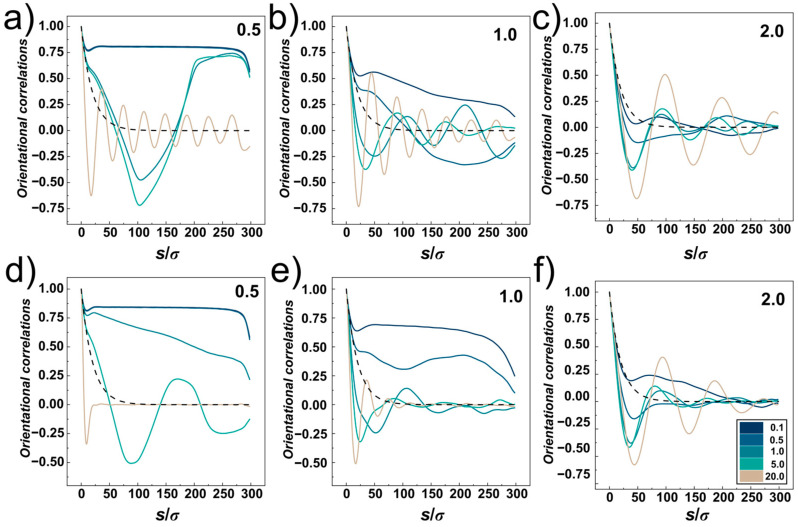
Orientational correlations as a function of the channel geometry, confinement strength and compressive force. Panels (**a**–**c**) show orientational correlations for cylindrical channels, and panels (**d**–**f**) show the orientational correlations obtained as average for right-handed and left-handed helical channels. The investigated range of confinement strengths *D*/*P* = 0.5, 1.0 and 2.0 is indicated by numbers in the upper right corner of the plots. The data are shown for compressive force *Fσ*/*ε*_0_ = 0.1, 0.5, 1, 5 and 20 indicated by the color-scale legend. The dashed line corresponds to the orientational correlations decay in the bulk, given as <cos*θ*> = exp (−*s*/*P*), where s is the coordinate along the chain.

**Figure 9 polymers-14-04201-f009:**
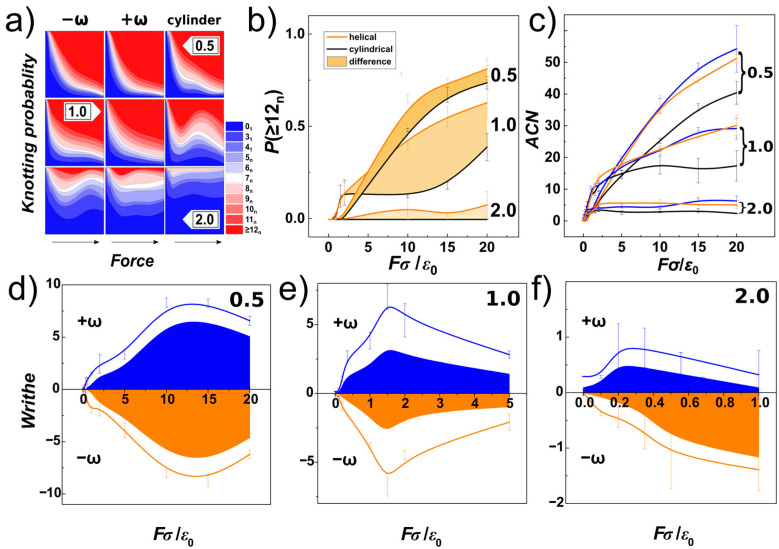
The topology of DNA chain and knots. (**a**) the graph shows knotting probability in terms of knot types and shown in thermometer colour scale representing knots by their crossing number and indicated by the legend. The knotting probability is shown for different confinement strengths expressed as *D*/*P* = 0.5, 1.0 and 2.0 (rows) and different geometries of the channel (columns). The comparison is made for right-handed (+ω) helical channels and left-handed (−ω) channels against cylindrical geometry. The arrow indicates direction of increasing compressive force in the range of *Fσ*/*ε*_0_ = <0;20>. (**b**) the graph shows difference of knotting probability (filled area) in channels with different geometry for knots with complex topology and crossing number above 11 and different confinement strengths *D*/*P* indicated by numbers adjacent to the curves. Orange lines correspond to averaged values for helical channels and blue lines correspond to the values obtained in cylindrical channels. (**c**) Average crossing number for different confinement strengths and geometries of the channel. Panels (**d**–**f**) show the dependence of writhe of the chain (lines) and writhe of the knotted portion (filled areas). The writhe of the chain and the knots obtained in the right-handed helical channels with positive handedness (+ω) is shown in blue and for the left-handed channels with negative handedness (−ω) is shown in orange. The writhe of the chain is investigated for three different confinement strengths indicated in terms of *D*/*P* = 0.5, 1.0 and 2.0 in the upper right corner of the plots.

## Data Availability

All data are available on request.

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
