# Peer review of "Knot Factories with Helical Geometry Enhance Knotting and Induce Handedness to Knots"

_polymers, 2022, doi:10.3390/polym14194201_

Round 1

Reviewer 1 Report

The authors study the conformation of DNA chains in cylindrical and helical nano-channels via computer simulations. They characterize the behaviour of the chains in absence and in presence of an external force and they further analyze the knotting probability and the handedness of the knots realized in the helical channels.

I find the paper surely worth of publication: the study is extensive and characterizes the system rather comprehensively.
I have a few comments that I suggest to consider:

1. The paper is quite long and touches various different aspect of the topic. In my opinion, the sections 3.1+3.2 could stand on their own and so could section 3.3. If I were the authors, I would consider splitting the paper in two.
2. I strongly recommend the authors to proof-read the manuscript carefully, there are a number of missing articles, misplaced commas and some small typos (such as "CAN" instead of "ACN" at line 682). Further, some sentences are not very clear, such as the one at line 286 "In such case, the polymer would..."
3. Instead of "are shown in thermometer scale" (line 610) please say "... are shown as a colormap".
4. The paper goes to great detail at times, in particular regarding the methodology (for example of knot detection). I would recommend, for the sake of readability, to cut these description to the essential, unless there is a methodological or scientific interest in the details (but then treat it in a separate section). In genereal, the paper is properly written but I suggest to try and shorten it, whenever possible.  
5. Please, I recommend to split the figures. There are currently 4 figures and each one has AT LEAST 6 panels. It is painful to go back and forth, sometimes of pages, to visualize what is written in the text.
6. At line 311, the figure 2c is misreferenced (should be 1c). Further, panels d, e and f of figure 4 lack lables and ticks on the x axis and the legend is unclear (it is also not straightforward from the text to understand it is the ratio D/P)

All in all, I recommend the paper in the present state for its content but I strongly suggest to improve the presentation as suggested above.

Author Response

Answers to Reviewer#1

The authors study the conformation of DNA chains in cylindrical and helical nano-channels via computer simulations. They characterize the behaviour of the chains in absence and in presence of an external force and they further analyze the knotting probability and the handedness of the knots realized in the helical channels.

I find the paper surely worth of publication: the study is extensive and characterizes the system rather comprehensively.

We thank the Reviewer for concise summary of our work and his/her conclusion.

I have a few comments that I suggest to consider:

1. The paper is quite long and touches various different aspect of the topic. In my opinion, the sections 3.1+3.2 could stand on their own and so could section 3.3. If I were the authors, I would consider splitting the paper in two.

We thank the Reviewer for this comment and generous suggestion that our work contains enough results for two papers. We are aware the manuscript contains extensive amount of data and in addition we did not show a lot of accessory data that did not make it to the manuscript but they are available on request. We would be glad to split the paper into a sequel of two stories, but we were generously offered by only one full waiver by the publisher. Thus, if we split the paper, the readers will be short of unravelling of the story before we get for them another open access option.

  1. I strongly recommend the authors to proof-read the manuscript carefully, there are a number of missing articles, misplaced commas and some small typos (such as "CAN" instead of "ACN" at line 682). Further, some sentences are not very clear, such as the one at line 286 "In such case, the polymer would..."

We thank the Reviewer for this comment. Based on the comment, we took the chance to go through the manuscript once again thoroughly and corrected the mis-formulations and typos. The changes are highlighted with the track changes option.

  1. Instead of "are shown in thermometer scale" (line 610) please say "... are shown as a colormap".

We thank the Reviewer for this comment as we were not sure if “thermometer” scale is a general term and we adopted it from the data processing software Microcal Origin Pro as a label for particular graph type. Based on the comment, we opted for the “colourmap” term.

  1. The paper goes to great detail at times, in particular regarding the methodology (for example of knot detection). I would recommend, for the sake of readability, to cut these description to the essential, unless there is a methodological or scientific interest in the details (but then treat it in a separate section). In genereal, the paper is properly written but I suggest to try and shorten it, whenever possible.  

We thank the Reviewer for this comment. We agree with the Reviewer, that technicalities are usually shared by the Supplementary Information attached to the main text of the paper or cut out completely and left on request perhaps. However, because of the specific case of the special issue that is aimed also on the “Development of new modeling and simulation techniques” and also because the solutions we provided in the manuscript on separating the writhe induced by confinement to the chain and to the knotted portion are not trivial and not by default included in the topology software, they are key to the story, together with the solution of the model of implicit confinement, we feel these should be kept in the main text of the manuscript.

  1. Please, I recommend to split the figures. There are currently 4 figures and each one has AT LEAST 6 panels. It is painful to go back and forth, sometimes of pages, to visualize what is written in the text.
    6. At line 311, the figure 2c is mis-referenced (should be 1c). Further, panels d, e and f of figure 4 lack lables and ticks on the x axis and the legend is unclear (it is also not straightforward from the text to understand it is the ratio D/P)

We thank the Reviewer for this comment. We did correct referencing to the figures in the text, but we also did split the composite figures into separate ones, and redistributed them across the text, so that they are more easily accessible without an excessive scrolling throughout the text. As for the last figure, originally Figure 4 (now Figure 9), we did corrections in the caption to the figure, but we feel we should keep the figure in its composite form, as we rely on the opinion of our mentors and experts on topology of DNA and knots, acknowledged by the end of the manuscript, who found the figure “absolutely excellent”.

All in all, I recommend the paper in the present state for its content but I strongly suggest to improve the presentation as suggested above.

We thank the Reviewer for the positive recommendation and also comments and suggestions aimed to improve the quality of the presentation. We went through the manuscript, we corrected referencing of the figures, mis-formulations and typos, redistributed the figures along the text in a more readable way and hopefully justified all of the Reviewer’s expectations.

Reviewer 2 Report

Referee report for the manuscript “Knot Factories with Helical Geometry Enhance Knotting and Induce Handedness to Knots” by Renáta Rusková and Dušan Račko

In this work the generation of knots in dna chains is explored by means of molecular dynamics simulations. The effects of confinement and compressive force are considered in both cylindrical and helical channels. The authors prove that the topology of the dna knots can be controlled by the negative or positive handedness of the helical channels. The analysis based on the molecular dynamics simulations is well described and sound. The results are discussed in detail and represent a new and interesting contribution for the community working of the chemical physics of macromolecules. For these reasons I suggest to accept the manuscript for publication on Polymers.

Author Response

Answers to Reviewer#2

In this work the generation of knots in dna chains is explored by means of molecular dynamics simulations. The effects of confinement and compressive force are considered in both cylindrical and helical channels. The authors prove that the topology of the dna knots can be controlled by the negative or positive handedness of the helical channels. The analysis based on the molecular dynamics simulations is well described and sound. The results are discussed in detail and represent a new and interesting contribution for the community working of the chemical physics of macromolecules. For these reasons I suggest to accept the manuscript for publication on Polymers.

We thank the Reviewer for the concise summary of our findings and recognising importance of our work for the community of the chemical physics of macromolecules. We did perform also English style and spell check as recommended by the Reviewer.